# Characteristics of Friction Plug Joints for AA2219-T87 FSW Welds

**DOI:** 10.3390/ma15041525

**Published:** 2022-02-18

**Authors:** Zhuanping Sun, Xinqi Yang, Shuxin Li

**Affiliations:** 1Tianjin Key Laboratory of Advanced Joining Technology, School of Materials Science and Engineering, Tianjin University, Tianjin 300350, China; xqyang@tju.edu.cn; 2Tianjin Long March Launch Vehicle Manufacturing Co., Ltd., Tianjin 300462, China; 3Capital Aerospace Machinery Co., Ltd., Beijing 100076, China; lishuxin2001@126.com

**Keywords:** AA2219-T87 aluminum alloy, fiction plug welding, plug and plate hole, mechanical property, microstructure, local strength

## Abstract

In this study, Friction plug welding (FPW) for 8 mm thickness AA2219-T87 sheets were carried out, and defect-free joints were obtained. The geometric size of plug and plate hole, rotational speed and welding force exhibit significant effects on the weld formation. Meanwhile, it is concluded that significant inhomogeneity of microstructure and mechanical properties exists in FPW joints. The recrystallization zone has the highest mechanical properties owing to the fine equiaxed grains and uniformly distributed θ precipitates. The entire plug, thermo-mechanically affected zone and nugget thermo-mechanically affected zone closed to the bonding interface are significantly softened due to the deformation of the grains and θ’ precipitate dissolution. The ultimate tensile strength (UTS) and elongation of the FPW joints can reach 359 MPa and 7.3% at 77 K and 305 MPa and 5% at 298 K, respectively.

## 1. Introduction

The 2219 aluminum alloy is one of the candidate materials for the rocket propellant storage tank owing to its high strength, low density, superior cryogenic property and excellent stress-corrosion resistance [1,2,3]. Friction stir welding (FSW) and bobbin tool friction stir welding (BTFSW) technologies have been widely used in the manufacture of rocket propellant storage tank. However, FSW and BTFSW will leave a terminal keyhole, which restricts their applications in circumferential weld structures. In order to fill the keyhole and repair other defects in FSW welds, a solid-phase FPW technology was developed by the welding institute [4,5]. The process is to use a high-speed rotating consumable plug under the action of axial force to squeeze into the hole prefabricated on the location where needs to be repaired. In FPW process, intense frictional heat generates on the interface of the plug and plate hole, which makes the metal around the contact interface reach thermoplastic state and realize metallurgical bonding [6,7]. This technology has the advantages such as low welding heat input, low residual stress and distortion, environmental protection, stable welding quality and high mechanical properties [8,9]. It can not only repair defects and keyholes of FSW welds but also welding of materials with poor weldability and harsh environments, such as welding of underwater petroleum pipelines [10]. Lockheed Martin applied FPW technology to repair the space shuttle outer tank made by 2219 and 2195 aluminum alloys in 2000 and the obtained repairing welds with high tensile strength, high fracture toughness and low defect rate, which greatly improved the production capacity of the space tank [11]. K Beamish studied the 10 mm thick AA6082-T6 FPW welding process and found that the welding force was 40 kN and the welding speed was about 1.7 m/s can get a high-quality joint. E. Dalder and others investigated the FPW of the “keyhole” of the AA2219 pressure vessel with an inner diameter of 1020 mm and a thickness of 38 mm and obtained a qualified weld [12,13,14].

The influences of plate hole and supporting plate hole structure and axial force on weld formation and tensile property of AA2219 FPW joints was studied [5]. However, the matching of the plug and plate hole, material flow behavior during the FPW welding process and the local mechanical properties of FPW joints have not been systematically clarified. In this paper, we will reveal the influence of plug and plate hole structure and welding parameters on weld formation and tensile properties of FPW joints. On this basis, the characteristics of microstructure, precipitates evolution, local tensile property and tensile properties at 298 K and 77 K of FPW joint for FSW weld have been analyzed and discussed systematically. This research will provide some important basis in repairing FSW defects for a rocket tank.

## 2. Experiments

The base material used in this study was 8 mm thick AA2219-T87 sheet; the dimensions of the plates are 300 mm × 150 mm × 8 mm. The measured 0.2% offset yield strength (YS), ultimate tensile strength (UTS) and elongation of AA2219-T87 plate are 387 MPa, 460 MPa and 12%, respectively. The measured UTS and elongation of FSW weld are about 390 MPa and 10%, respectively. The plug material was AA2219-T6. The nominal chemical composition of AA2219-T87 sheet and 2219-T6 plug are listed in Table 1. All the FPW experiments were conducted on the FPW equipment developed by the welding laboratory of Tianjin University. In order to study the influence of plug and plate hole on the weld formation of FPW joint, four kinds of plug and plate hole combinations were designed; the plug and plate hole were designed into conical shape to improve the frictional heat generation and compared with straight plate hole. At the same time, in order to prevent the plug and plate hole from being stuck during the FPW process, the taper angle of the plug is 5° smaller than that of the plate hole, as shown in Figure 1. The welding parameters used in the experiment are listed in Table 2. 

The FPW process is that the high-speed rotating plug moves toward the plate hole at a certain speed (as shown in Figure 2a), and the initial contact locates at the bottom, where frictional heating and plastic deformation occur between the plug and plate hole (as shown in Figure 2b). As the welding process continues, contact area between the plug and plate hole gradually increases and the friction heat generation also increases continuously. Then, full contact of the plug and plate hole and the bonding interface reach thermoplastic state (as shown in Figure 2c). When the plug reaches the set consumption, the high-speed rotating plug stops abruptly and applies forging force to refine grains (as shown in Figure 2d).

FPW joints were cut along the diameter of the plug for macro- and micro-observation and hardness test and the cross-sections were ground with abrasive paper, mechanically polished and etched with corrosive (2 mL HF + 3 mL HCl + 5 mL HNO_3_ + 190 mL H_2_O) for 15 s and then observed under optical Olympus GX51 optical microscope. The hardness distribution was measured by HVS-1000 micro-hardness tester in 1 mm steps using 1 kg load and a dwell time of 10 s. The tensile tests were conducted on the CSS-44100 universal testing machine with a loading rate of 3 mm/min. The dimension of tensile specimens is given in Figure 3b.

In order to further study the FPW joint for FSW weld (as shown in Figure 2a), the as-welded specimens were cut along the parallel and perpendicular direction of FSW welds for microstructure observation, precipitates and electron backscatter diffraction (EBSD) analysis. The Tecnai G2 F20 transmission electron microscope (TEM) was used to characterize the precipitates at different zones of FPW joint. EBSD analysis was carried out using FEL SciosDualBeam with HKL Channel 5 EBSD probe. To examine the tensile properties of different welded zones, micro-tensile test is then performed on Instron micro-test system RDL50 with an electronic extensometer. The micro-tensile samples were extracted at the same locations corresponding to the microstructure observation. The dimension of micro-tensile specimens is given in Figure 3d. Considering the service environment of the rocket tank, both the room (298 K) and low temperature (77 K) tensile properties of the FPW joints were tested on the CSS-44100 universal testing machine with a loading rate of 3 mm/min; the dimensions of low-temperature tensile specimens are given in Figure 3c. All these tests and analyses were completed within 7–10 days of FPW welding.

## 3. Results and Discussion

### 3.1. Plug and Plate Hole Structure

The joint with four kinds of plug and plate hole combinations in Figure 1 was welded using the D4 parameter in Table 2. Figure 4 shows the cross-sectional appearances and defects of FPW joints. Type A and Type C use the same plug, and the diameter of the top plate hole is also the same, except for the difference between the straight hole and the taper hole. Unwelded defects are prone to form at the bonding interface of the Type A joints owing to the poor frictional heat. Defect-free weld was obtained at Type C joint, which is related to the larger contact area of the plug and taper plate hole compared with that of Type A straight plate hole. Weak-bonding defect and tissue over-burning defect are observed in both the top and bottom part at the bonding interface of Type B joint, respectively. The weak-bonding defects are caused by the insufficient normal force perpendicular to the bonding interface due to the decrease of plug and plate hole angle. The tissue overheating defects are caused by the relatively higher frictional heat owing to the decrease of diameter at the bottom of the plug and plate hole. Plug-breakage defects exist at the Type D joint, which were characterized by the separation of fully plasticized bottom material and unplasticized top material of the plug owing to larger difference in linear velocity and temperature gradient between top and bottom of the FPW joint. Therefore, the following FPW experiments were carried out based on Type C combination of plug and plate hole.

### 3.2. Weld Formation

Figure 5 shows the cross-sectional appearance of FPW joints. The weld quality is shown in Table 3. Figure 6 shows the welding defect morphology of regions marked in Figure 5. It can be seen that all the FPW joints have five types of defects, which are unwelded defect, weak-bonding defect, hole-type defect, tissue loose defect and insufficient filling defect. In FPW process, the rotational speed and welding force are the most important parameters affecting weld formation, which directly determine the frictional heat generation at the bonding interface. The welding force can be divided into tangential force Ff along the bonding interface and normal force Fn perpendicular to the bonding interface as marked in the FPW joint of Figure 5 at rotational speed 5500 rpm and welding force 45 kN. When the rotational speed and welding force are lower than 5500 rpm and 40 kN, defect-free joints can not be achieved and are prone to generate unwelded defects, hole-type defect and weak-bonding defect with the magnified observations given in Figure 6a,b. When the rotational speed is 4500 rpm, the diameter of the plug is thicker than that of the original plug under the selected pressure. When the rotational speed is 5500 rpm, the plug greatly deformed compared with that of 4500 rpm, as the welding force increases to 45 kN. The plug severely deformed and “necked down” near the bottom surface which is caused by the uneven normal force acting on the bonding interface. D_h_ respects the distance from the necking start position to the bottom surface, as marked in the FPW joint of Figure 5. At rotational speed 5500 rpm and welding force 45 kN, generally speaking, the lower value of D_h_ usually means sufficient frictional heat and good material flow capability. This indicates that the increase in rotational speed is more effective in increasing frictional heat generation than welding force. When the rotational speed is higher than 6500 rpm, D_h_ decreases significantly with increasing rotational speed and welding force. Moreover, the defects are significantly reduced. Weak-bonding defect and tissue loose defect (as shown in Figure 6d,e) exist only under the welding force of 30 kN. However, it is not certain that a defect-free joint can be obtained if the rotational speed and welding force are large enough. When the rotational speed is 7500 rpm, lack of filling is found due to insufficiency of the pushing amount (as shown in Figure 6f). It is concluded that a high-quality joint is determined by the match of rotational speed, welding force, forging force and pushing amount.

### 3.3. Mechanical Properties

#### 3.3.1. Hardness Distribution

Figure 7 shows the hardness of FPW joints at the welding parameters of 40 kN and 4500–7500 rpm. Three lines were measured for each FPW joint as top, middle and bottom, which are 1 mm, 4 mm and 7 mm from the top surface. It can be seen that the hardness of the plug is severely softened at all welding parameters ranging from 80 to 100 HV. Combined with the cooling rate of bonding interface at both plug side and base metal side during FPW process of 8-mm thick AA 2219-T87 revealed by B Du et al. (2020), the peak temperature of the plug side is about 70 °C higher compared with that of the base metal side, but the cooling rate ranging from 53.3 to 38.2 °C/s is significantly lower than that of the plug side ranging from 80.3 to 58.5 °C/s from the top surface to bottom [4]. In addition, the diameter of the plug is thinner than that of base metal and closed environment from the plate hole and supporting plate, which are unfavorable to heat dissipation. The base metal has the highest hardness ranging from 135 to 150 HV. The middle hardness for 7500 rpm higher in the distance 5–15 mm from the center of the plug (as shown in Figure 7b), which is related to the longer friction time and slower heat dissipation. The higher frictional heat enables the plastic metal to flow and recrystallize sufficiently, and the precipitates is also more abundant. However, due to the longest friction time in the bottom (as shown in Figure 7c), the larger frictional heat for 7500 rpm results in grain growth and hardness reduction.

#### 3.3.2. Tensile Properties

Figure 8 shows the tensile property of A1–D5 joints; for each group of welding parameters, three specimens were tested to get an average value. It is clearly seen that rotational speed has an important effect on the tensile property of FPW joints. When the rotational speed is 4500 rpm, the UTS and elongation of the FPW joints are greatly decreased compared with that of other rotational speeds, and the UTS are lower than 280 MPa. Combined with the result in Figure 5, it is concluded that rotational speeds above 5500 rpm are prone to obtain high quality FPW joints. Welding force also has a great effect on the UTS and elongation of FPW joints; when rotational speeds are 4500 and 5500 rpm, the UTS and elongation continue increasing as the welding force ranges from 30 to 50 kN. When the rotational speeds are 6500 and 7500 rpm, the UTS and elongation first increase and then decrease. The maximum UTS (363 MPa) and elongation (8.0%) are obtained at welding parameters of 7500 rpm and 40 kN and 7500 rpm and 45 kN, which are close to that of FSW joint, which reached 78.9% and 67% of base metal, respectively.

The fracture positions of the tensile specimens are shown in Table 4 and Figure 9. The typical fracture positions of FPW joints are seen. When the rotational speed is 4500 rpm, the fracture positions of the joints obtained in all the welding forces are in the bonding interface. This is due to the small heat input; the plug and base metal cannot fully reach the thermoplastic state, and the effective connection cannot be achieved. When the rotational speed exceeds 5500 rpm, with the increase of welding force, the fracture position transitions from the bonding interface to the thermo-mechanically affected zone. When the rotational speed exceeds 6500 rpm, most of the fracture positions are in the thermo-mechanically affected zone.

The results indicate that the rotational speed exerts a significant effect on weld formation and tensile property, followed by the welding force. It is finally determined that the 8 mm thick AA2219-T87 FPW parameters are: 6500 rpm and 40–50 kN and 7500 rpm and 35–45 kN.

### 3.4. FPW Joint for FSW Weld

#### 3.4.1. Material Flow 

FPW for FSW welds were performed using Type C plug and plate hole combination and D4 welding parameters due to defect-free weld formation and higher mechanical properties. Figure 10 and Figure 11 illustrate the material flow direction of FPW joints along the parallel and perpendicular directions of FSW welds, respectively. It is obvious that there exists a demarcation point in the base metal, where the direction of material flow changes (as shown in Figure 10a–c). The plastic material on both sides of the bonding interface flows to the top and bottom surfaces of the FPW joint to form flashes under the action of the plug rotation and welding pressure during FPW process. Material flows on the plug mainly along the axial direction; the plastic deformation streamlines at the top of the plug are dispersed and the grains are relatively large and compressed in the radial direction (as shown in Figure 10d). However, the plastic deformation streamlines at the bottom of the plug are concentrated and the grains are narrowed in radial direction and significantly elongated in axial direction due to strongly restrained by the plate hole (as shown in Figure 10e). Material flow in the FSW nugget is not obvious. However, from the flash of the top and bottom, the flow direction of plastic materials should change at a certain demarcation point (as shown in Figure 11a,b).

The open deformation feature of the plastic material at the base metal and nugget expands the contact area with the plug, which is beneficial to the formation of FPW joint and improvement of the connection strength.

#### 3.4.2. Microstructures

The cross-sectional microstucture of FPW joint along the parallel and perpendicular direction of FSW weld are shown in Figure 12 and Figure 13. The whole joint consists of six zones: recrystallized zone (RZ), plug thermo-mechanically affected zone (PTMAZ), plug heat-affected zone (PHAZ), nugget thermo-mechanically affected zone (NTMAZ)/base metal thermo-mechanically affected zone (TMAZ), nugget heat-affected zone (NHAZ)/base metal heat-affected zone (HAZ) and nugget zone (NZ)/base metal (BM). The RZ is located at the bonding interface, where the material is plasticized and dynamic recrystallization occurs owing to the action of strong frictional heat and welding force, then the recrystallized grains formed equiaxed grains under the action of forging force. The width of the RZ ranged from 15 to 60 μm and gradually increases from the top surface to the bottom surface along the thickness of FPW joint due to different friction time and heat dissipation condition (as shown in Figure 12a–c and Figure 13a–c). In PTMAZ, NTMAZ and TMAZ close to the bonding interface, the grains are coarsened and deformed compared with that of plug, nugget and base metal under the action of drastic friction thermal and mechanical force (as shown in Figure 12d,g and Figure 13d,g). In PHAZ, NHAZ and HAZ relatively far from the bonding interface, the microstructure was only affected by frictional heat; the grains retained the characteristics of the nugget/base material and plug (as shown in Figure 12e,h and Figure 13e,h).

#### 3.4.3. Precipitate Evolution

Figure 14 shows the TEM observations of different zones in the FPW joint for FSW weld. AA2219-T87 is a heat-treatable strengthened aluminum alloy; the evolution of that precipitated during welding is one of the main factors affecting the performance of the FPW joint. In BM (Figure 14c), a large amount of fine θ’ precipitates are distributed in grains and grain boundaries, referring to Figure 14K. In TMAZ and HAZ (Figure 14a,b,i), the amount of θ’ precipitates are decreased significantly and grew slightly compared with BM; moreover, θ precipitates with larger diameters can be observed referring to Figure 14l. In RZ (Figure 14j), there are a large number of uniformly distributed θ precipitates with smaller diameters compared to that of TMAZ and HAZ; however, the θ’ precipitates dissolved almost completely. As shown in Figure 14d–f, the number of θ precipitates in NTMAZ and NHAZ decreased obviously compared with NZ, which is related to dynamic recrystallization. From Figure 14d–f, it was found that the entire plug is affected by frictional heat, the amount of θ’ precipitates in PTMAZ and plug center decreased obviously and there exist θ precipitates with larger diameters compared with original plug.

The results indicate that a large number of original θ’ precipitates dissolved during welding process and some of the θ’ precipitates transformed to θ particles under very high frictional heat and severe plastic deformation during the cooling and aging process.

The EBSD analysis results are shown in Figure 15. According to the IPF coloring (Figure 15a,d,g,j), the grains in HAZ and PHAZ severely deformed and recrystallized grains appeared along the grain boundary in PHAZ caused by strong frictional heat effects and extruding during FPW welding process. According to the misorientation angle analysis results (Figure 15b,e,h,k), the distribution characteristics of the grain orientation angle in HAZ and PHAZ are similar, mainly concentrated within 10°. The grain orientation angle in RZ is uniformly distributed from 0 to 60°. This indicates that the grains in HAZ and PHAZ have obvious preferred orientation. NHAZ, dominated by large-angle grains accompanied by grain orientation angle within 10° due to the original microstructure in nugget, consists of equiaxed grains, which should be large-angle grain boundaries; however, the equiaxed crystals close to the bonding interface are strongly affected by frictional heat and mechanical force during the FPW process, making the equiaxed crystals tend to deform direction. The misorientation angle affects the deformation and fracture behavior of the material: the larger the misorientation angle, the greater the interfacial energy, which can effectively hinder the propagation of cracks and is conducive to improving the mechanical properties of FPW joints [15]. According to the grain size analysis results (Figure 15c,f,i,l), in RZ and NHAZ, grain diameters concentrated in 2–15 µm. In HAZ and PHAZ, grain with diameters ranged from 2 to 60 µm approximately.

#### 3.4.4. Local Tensile Property

Figure 16 shows the YS, UTS and elongations of the specimens extracted from different zones of the FPW joint for FSW weld micro-tensile test. In RZ, the YS and UTS are 247 and 366 MPa, respectively, reached 63.8% and 79.5% equivalent to that of the base metal, and the elongation is 14.5%, reached 121% equivalent to that of the base metal. In TMAZ, HAZ, NTMAZ and NHAZ, the YS, UTS and elongation are gradually decreased with the decrease of the distance to bonding interface. In PTMAZ and PHAZ, the YS, UTS and elongation are lower than other regions owing to the entire plug being affected by strong frictional heat and mechanical force; the PTMAZ has the minimum YS (193 MPa) and UTS (267 MPa) and elongation (5.2%), respectively, reached 49.9%, 58% and 43% equivalent to that of the base metal.

#### 3.4.5. Tensile Property

Figure 17 shows the tensile properties of the base metal, FSW joints and FPW joints for FSW weld at 298 K and 77 K, respectively. FPW joints were obtained at D4 parameters and three tensile specimens were tested for each type of joint. The results show that the UTS and elongation of the base metal, FSW joints and FPW joint at 77 K are significantly higher compared with that of 298 K. The UTS and elongation of the FPW joints at 77 K are 359 MPa and 7.3%, respectively, which is 17.7% and 32.7% higher than those at 298 K and reached 96% and 73% equivalent to that of the FSW joints at 77 K. This indicates that FPW have obvious advantages in the repair of FSW welds.

#### 3.4.6. Discussion

According to the analysis results of the hardness distribution, local tensile property and mechanical properties of the FPW joints, it can be seen that the local tensile property and mechanical properties are both lower than that of the base metal owing to the inhomogeneity of the microstructure, grain size and precipitates in different regions. The FPW joint undergoes complex processes such as recrystallization, grain deformation and growth and evolution of precipitates under the very high temperature and strong mechanical force, which inevitably affect the mechanical properties. The mechanical properties of FPW joints are discussed as follows:

Dynamic recrystallization occurs in RZ, a large volume of θ’ phase dissolved and the initial grains transformed to fine equiaxed grains. According to Hall–Patch formula $\sigma_{d}=\sigma_{0}+K_{d}d^{-1/2}$, where $\sigma_{0}$ and $K_{d}$ are constants, and it can be seen that the yield strength $\sigma_{d}$ is inversely proportional to the grain diameter. Moreover, sufficient grain boundaries also increase the resistance to plastic deformation [16,17]. These are all the methods of improving YS, UTS and elongation. However, it is difficult to significantly improve the mechanical properties of the FPW joints due to the narrow width that ranged between 15 and 60 μm.

The TMAZ and PMAZ are the weakest softening regions of FPW joints owing to the severe grain deformation and growth and massive θ’ precipitates dissolution, where the crack generates and then fractures areas of FPW joints with well-bonded friction interface. Rotational speed and welding force are the most important factors affecting the welding heat input during FPW process. As the rotational speed increases from 4500 to 6500 rpm, the UTS significantly improves ranging from 20 to 160 MPa under the welding force of 30–50 kN. However, the increase in welding force from 40 to 50 kN brings a slight increase in UTS ranging from 7 to 30 MPa. Meanwhile, the increase of rotational speed brings obvious plug deformation and large flash under the same pressure. However, the welding force increased to 50 kN still cannot effectively improve the heat input, making the friction interface completely reach thermoplastic state to achieve effective connection under the rotational speed of 4500 rpm. Therefore, the contribution of rotational speed to the improvement of mechanical properties is greater than that of welding force.

## 4. Conclusions

FPW experiments were performed on AA2219-T87 sheet with 8 mm in thickness, and defect-free joints were obtained under Type C plug and plate hole combination and optimized parameters. The geometric size of the plug and plug hole, rotational speed and welding force exert significant effect on weld formation and tensile properties.

Plastic metal of the plug and base metal on both sides of the bonding interface flow to the top and bottom surfaces of the FPW joint to form flashes at a certain demarcation point under the action of the plug rotation and welding force during FPW process, which is the key for the weld formation.

The FPW joint has obvious microstructure inhomogeneity, which can be divided into six regions: RZ, PTMAZ, PHAZ, TMAZ, HAZ and BM. RZ undergoes dynamic recrystallization to form fine equiaxed crystals with width ranging from 15 μm to 60 μm and gradually widens from top to bottom along the thickness of FPW joint due to different friction time and heat dissipation condition.

Local softening was found in PTMAZ and TMAZ of FPW joint closed to the bonding interface owing to severe grain deformation and growth and massive θ’ precipitates dissolution, which exert an important effect on mechanical properties. The UTS and elongation of the FPW joints can reach 359 MPa and 7.3% at 77 K and 305 MPa and 5% at 298 K, respectively.

## Figures and Tables

**Figure 1 materials-15-01525-f001:**
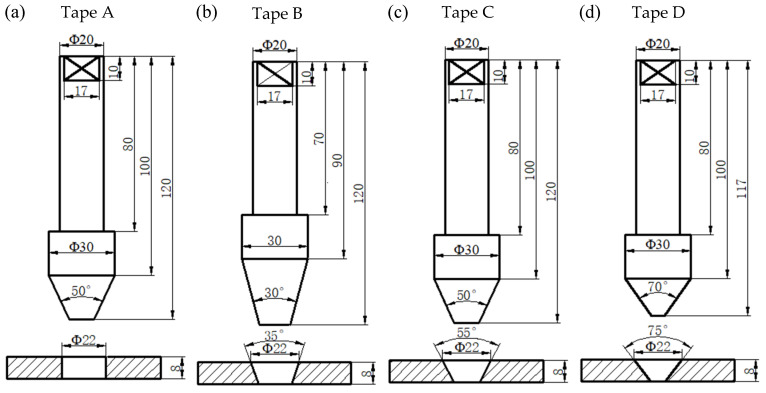
Geometric dimensions of plug and plate hole (mm): (**a**) 30° plug and straight plate hole, (**b**) 30° plug and 35° plate hole, (**c**) 50° plug and 55° plate hole and (**d**) 70° plug and 75° plate hole.

**Figure 2 materials-15-01525-f002:**
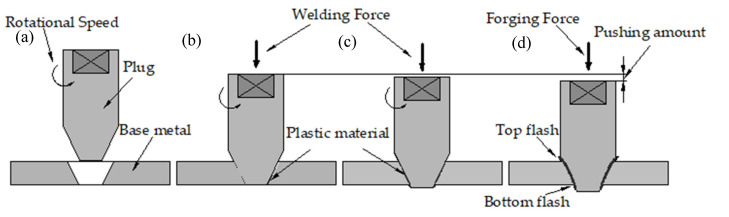
Schematic illustration of FPW process: (**a**) the plug move toward the plate hole, (**b**) the plug contact the bottom of the plate hole, (**c**) the plug completely contact the plate hole and (**d**) forging.

**Figure 3 materials-15-01525-f003:**
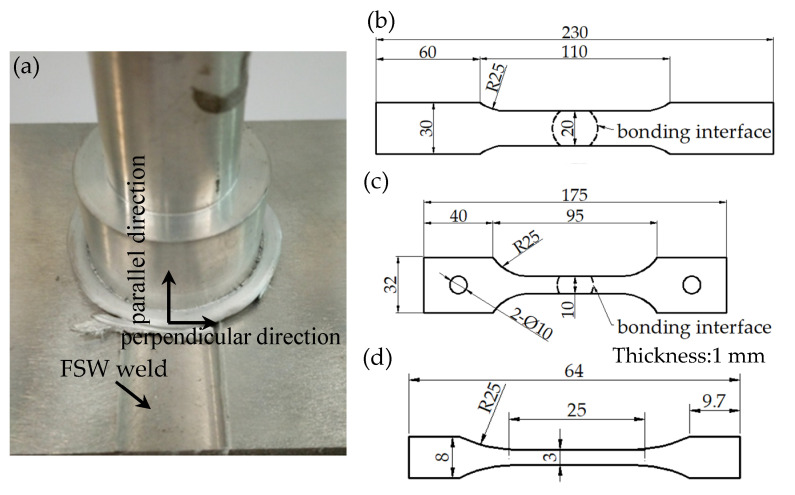
FPW joint for FSW weld and geometric dimensions of tensile specimens (mm): (**a**) FPW joint for FSW weld, (**b**) room temperature tensile specimen, (**c**) low temperature tensile specimen, and (**d**) micro-tensile specimen.

**Figure 4 materials-15-01525-f004:**
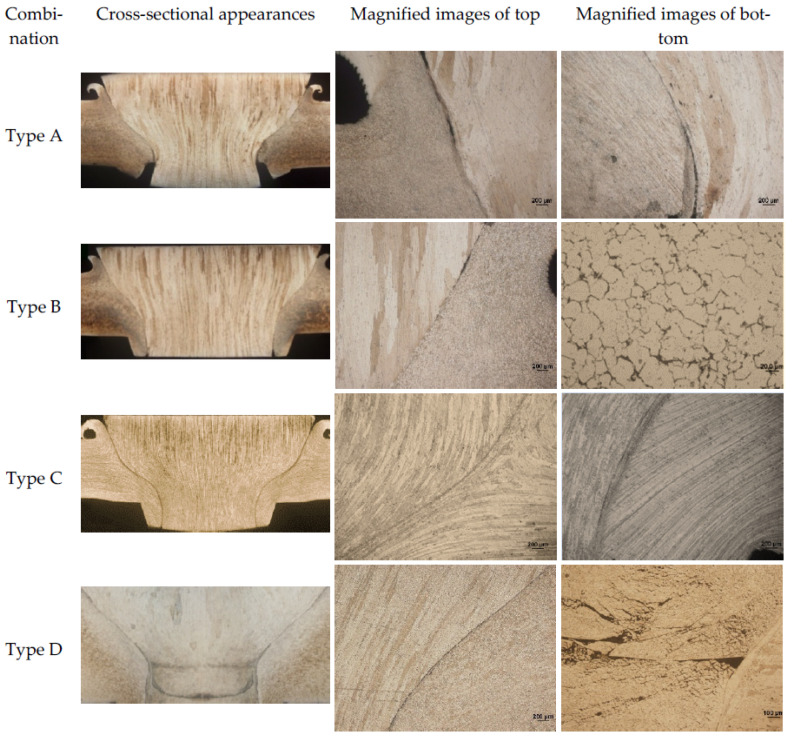
Macroappearance and defects of FPW joint.

**Figure 5 materials-15-01525-f005:**
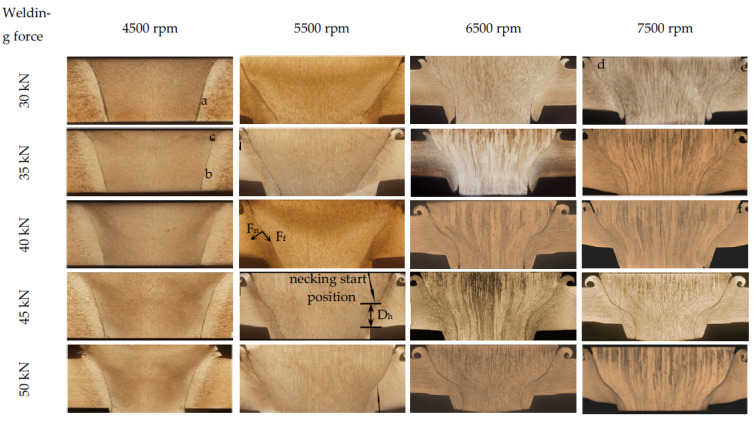
Cross-sectional appearance of FPW joints.

**Figure 6 materials-15-01525-f006:**
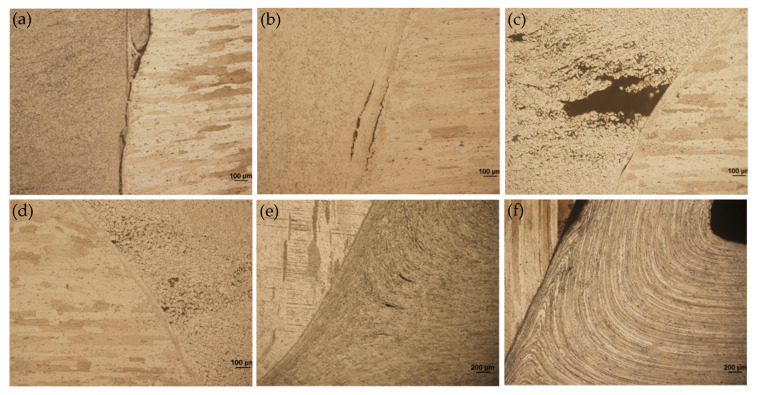
Welding defect morphology of regions marked in Figure 5: (**a**) region a, (**b**) region b, (**c**) region c, (**d**) region d (**e**) region e and (**f**) region f.

**Figure 7 materials-15-01525-f007:**
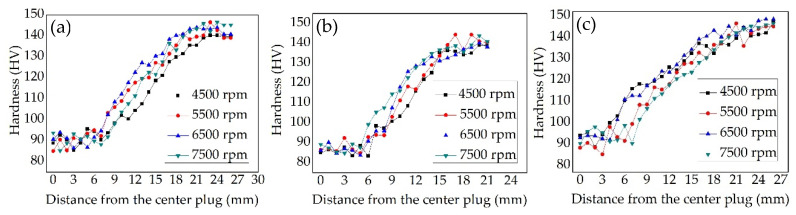
Hardness distribution of FPW joints: (**a**) top, (**b**) middle and (**c**) bottom.

**Figure 8 materials-15-01525-f008:**
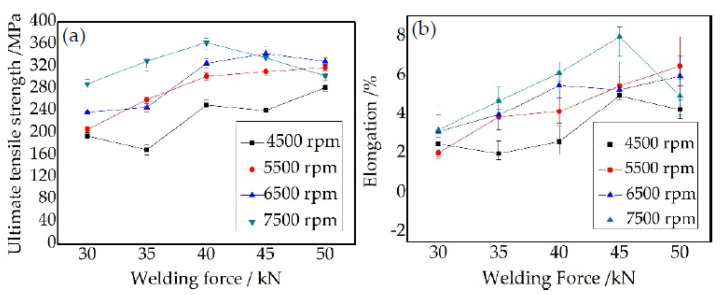
Tensile property of A1–D5 joints: (**a**) UTS and (**b**) elongation.

**Figure 9 materials-15-01525-f009:**
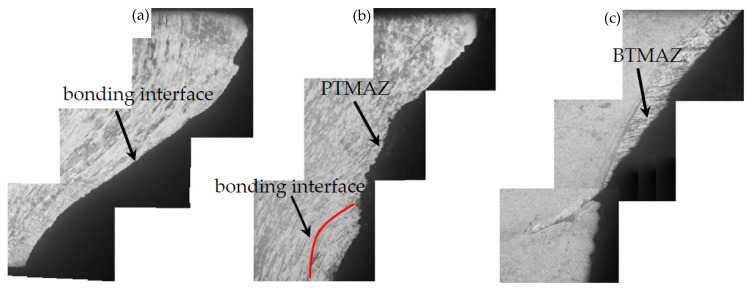
Fracture positions: (**a**) D1 joint, (**b**) D3 joint and (**c**) D4 joint.

**Figure 10 materials-15-01525-f010:**
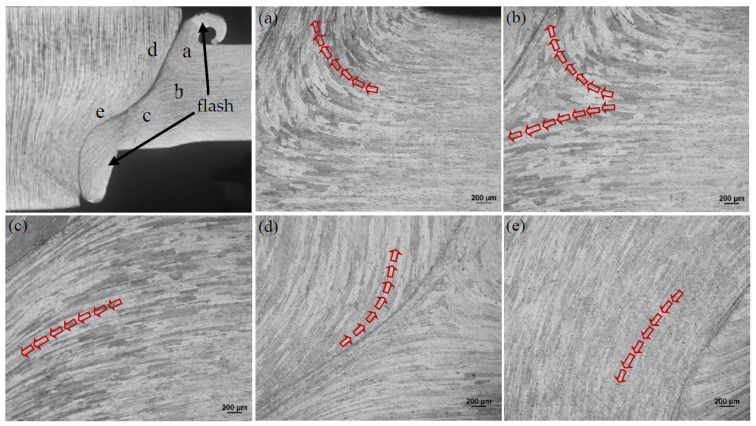
Material flow direction of FPW joint along the perpendicular direction of FSW: (**a**) top of the base metal, (**b**) middle of the base metal, (**c**) bottom of the base metal, (**d**) top of the plug and (**e**) bottom of the plug.

**Figure 11 materials-15-01525-f011:**
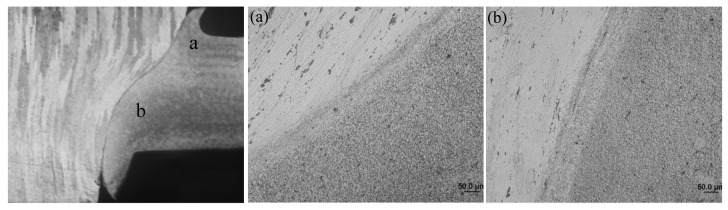
Material flow direction of FPW joint along the parallel direction of FSW: (**a**) top of the nugget, and (**b**) bottom of the nugget.

**Figure 12 materials-15-01525-f012:**
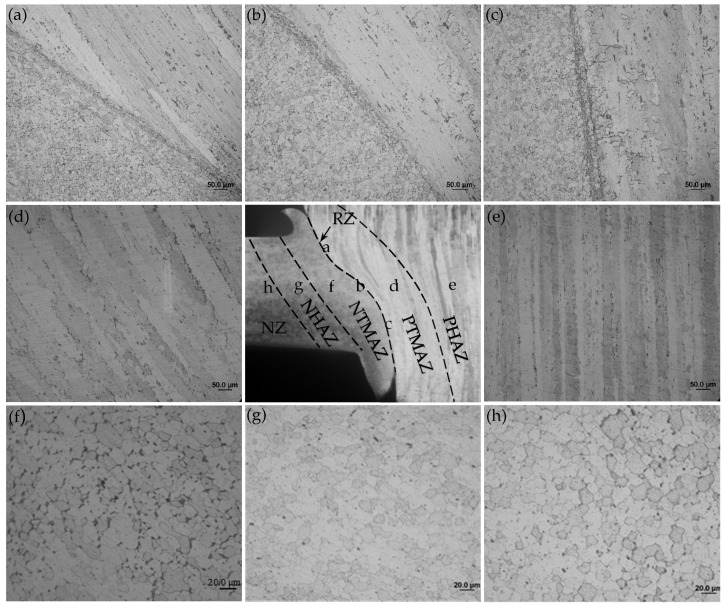
Microstructures of D4 joint along the parallel direction of FSW weld: (**a**) top RZ, (**b**) middle RZ, (**c**) bottom RZ, (**d**) PTMAZ, (**e**) PHAZ, (**f**) NTMAZ, (**g**) NHAZ and (**h**) NZ.

**Figure 13 materials-15-01525-f013:**
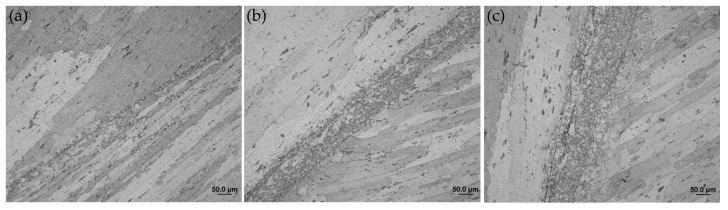
Microstructures of D4 joint along the perpendicular direction of FSW weld: (**a**) top RZ, (**b**) middle RZ, (**c**) bottom RZ, (**d**) PTMAZ, (**e**) PHAZ, (**f**) NTMAZ, (**g**) NHAZ and (**h**) NZ.

**Figure 14 materials-15-01525-f014:**
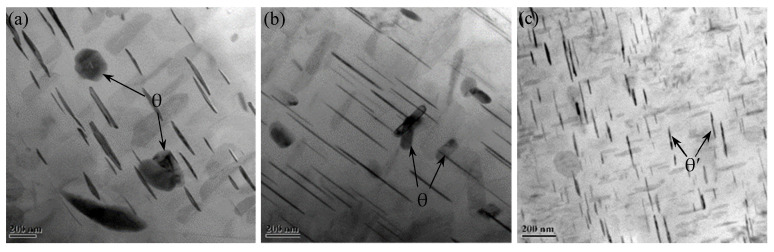
Morphology and distribution of the precipitates: (**a**) TMAZ, (**b**) HAZ, (**c**) BM, (**d**) NTMAZ, (**e**) NHAZ, (**f**) NZ, (**g**) PTMAZ, (**h**) plug center, (**i**) original plug, (**j**) RZ, and electron diffraction pattern of (**k**) θ’ precipitates in BM, and (**l**) θ precipitates in HAZ. 3.4.4 EBSD analysis.

**Figure 15 materials-15-01525-f015:**
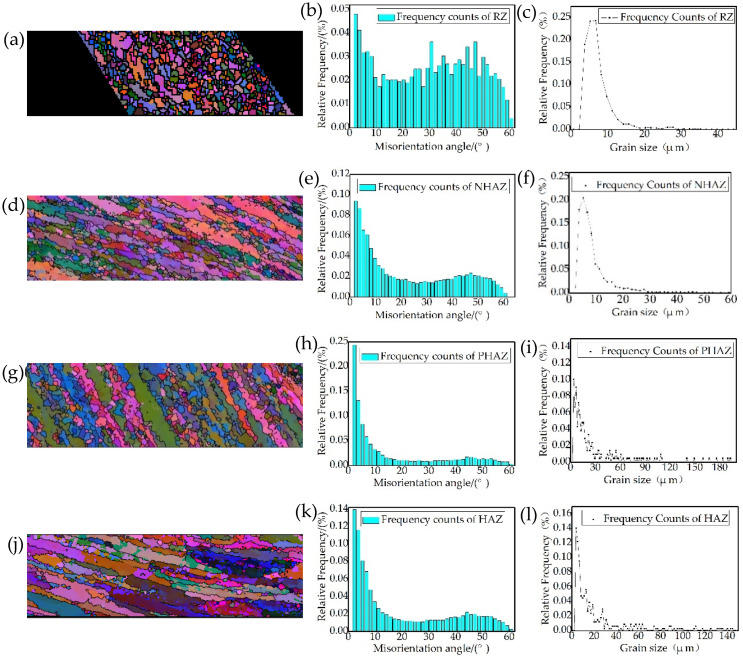
EBSD analysis of FPW joint for FSW weld: (**a**) RZ IPF coloring, (**b**) RZ misorientation angle, (**c**) RZ grain size, (**d**) NHAZ IPF coloring (**e**) NHAZ misorientation angle (**f**) NHAZ grain size (**g**) PHAZ IPF coloring (**h**) PHAZ misorientation angle (**i**) PHAZ grain size, (**j**) HAZ IPF coloring (**k**) HAZ misorientation angle and (**l**) HAZ grain size.

**Figure 16 materials-15-01525-f016:**
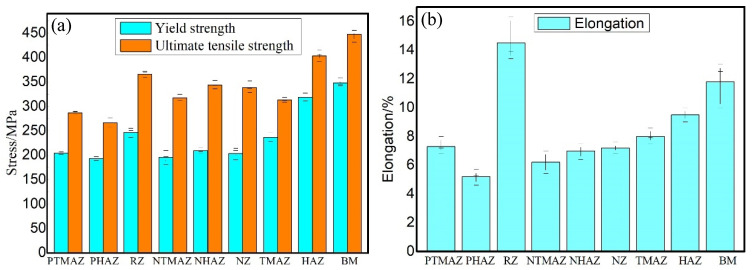
Tensile properties of different zones of the joints: (**a**) YS and UTS and (**b**) elongation.

**Figure 17 materials-15-01525-f017:**
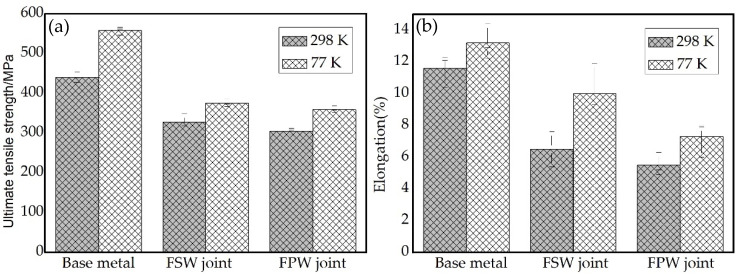
Tensile property of bass metal, FSW joints and FPW joints at 298 K and 77 K: (**a**) UTS and (**b**) elongation.

**Table 1 materials-15-01525-t001:** Chemical composition of AA 2219 in wt%.

Element	Cu	Mn	Si	Fe	Mg	Zn	Ti	Zr	V	Al
AA2219-T87	6.20	0.30	0.07	0.15	0.02	0.03	0.06	0.14	0.10	Balance
AA2219-T6	6.33	0.32	0.06	0.23	0.01	0.04	0.06	0.2	0.08	Balance

**Table 2 materials-15-01525-t002:** Welding parameters and specimen no.

Specimen No.	Rotational Speed (rpm)	Welding Force (kN)	Forging Force (kN)	Pushing Amount (mm)	Welding Speed (mm/min)
A1		30	35	8	50
A2		35	40	8	50
A3	4500	40	45	8	50
A4		45	50	8	50
A5		50	55	8	50
B1		30	35	8	50
B2		35	40	8	50
B3	5500	40	45	8	50
B4		45	50	8	50
B5		50	55	8	50
C1		30	35	8	50
C2		35	40	8	50
C3	6500	40	45	8	50
C4		45	50	8	50
C5		50	55	8	50
D1		30	35	8	50
D2		35	40	8	50
D3	7500	40	45	8	50
D4		45	50	10	50
D5		50	55	10	50

**Table 3 materials-15-01525-t003:** Weld quality of FPW joints.

Welding Force (kN)	Rotational Speed (rpm)			
	4500	5500	6500	7500
30	▲	▲	▲	◆
35	★●	★	-	-
40	★	★	-	■
45	★	★	-	-
50	★	-	-	-

▲ unwelded defect, ★ weak-bonding defect, ● hole-type defect, ◆ tissue loose defect, ■ insufficient filling defect.

**Table 4 materials-15-01525-t004:** Fracture location of FPW joints.

Rotational Speed (rpm)	Welding Force (kN)				
	30	35	40	45	50
4500	▲	▲	▲	▲	▲
5500	▲	▲	▲	◆	●
6500	▲	●	◆	●	◆
7500	▲	▲	●	◆	◆

▲ bonding interface, ◆ base metal thermo-mechanically affected zone (BTMAZ), ● plug thermo-mechanically affected zone (PTMAZ).

## Data Availability

Not applicable.

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
