# Peer review of "Characteristics of Friction Plug Joints for AA2219-T87 FSW Welds"

_materials, 2022, doi:10.3390/ma15041525_

Round 1
Reviewer 1 Report
The manuscript aims at presenting the processing features of a FPW process on the 2219-T87 alloy incruding material microstructure changes. However, the article shows its very low weakness starting from the title, then following the Abstract, the Introduction and the scientific sections. Namely, publishing a scientific paper on the "characteristics" of a process sounds quite unattractive. Besides all, the most discouraging feature of the article is its poor English form which inhibits the required understanding of the authors' research and findings. Moreover, they often use inappropriate technical terms, and state materials science considerations mistakedly.
It follows some examples of critical issues, knowing that the followimg list is far to be complete. For instance, inappropriate metallurgical terms are a. "thermoplastic" (typically used for polymers) rather than thermomechanical
plasticity, b. connection rather than bond or welded joint, c. "materials" in spite of the fact that the manuscripts deals with one single alloy material, d. "crystal grains are compressed", rather than "grains are plastically strained in compression, e. "Al2Cu" precipitate should read 2 as an index, and the like. As per the comments on the manuscript.
The Abstract includes a testionable metalurgical statements: " greater residual stress and distortion will occur, and the joint strength will also be greatly reduced..." In fact, the strength should not be reduced, rather toughness will be reduced. In the Introduction section quotes to previous studies is made but references are not included or wrongly referrred to. In the text, several sentences are abosolutely incomprehensible, often due to confusing English grammar and punctuation.
Thus, the manuscript in the current form cannot be reviewed due to very severe lacking of English clarity. Thus the manuscript cannot fullfil even the minimal requirements for a sound review.
The following is a summary of other important lacking points found which may be useful to the authors to improve in the case they are intended to resubmit their manuscript.
In general, the dominant part of the article is of the technological relevance rather than materials science one. If the article will be resubmitted to Materials journal such proportions should should be reverted to give more relevance the material aspects rather than the technological aspects. With this respect, information on the precipitate morphology and grain size of the base material must be given before for a fair comparison with the microstructure after FPW.
1 Notation of the relevant zones in the FPW region, as described in figure 12, should be anticipated; such acronyms are used before in the text but not defined yet.
2 The meaning of the transitional arc is not defined, hence not well understood
3 Throghout the paper the notations of "upper" and "lower" regions should be better replaced with "top" and "bottom" regions respectively
4 Figure 7: missing units in the legends;
5 Figures 7,8: discussion and interpretation of the curves is inappropriate
6 Figure 14: the specification that the image comes from TEM analysis must be given
7 Specification of the time passed after FPW before tensile testing and TEM inspection must be given
8 Estimates on the cooling rate after FPW must be provided along with the orientative time cycle of FPW.
9 It is highly disputable the significant role ascribed to "precipitate evolution" without a fair comparison between precipitates morfology before and after FPW.
10 On the following statament:
"In RZ, it can be found that although a large volume of θ″ phase dissolved, the initial grains transform to fine equiaxed grains and generated a large number of dislocations." is highly questionable unless clear experimental proof is given. Moreover, it is unsound that the pecipitate dissolution during severe heating in FPW is less important but overaging after welding and cooling is dominant.
Author Response
Point 1: I found that sections 2 & 3 should be re‐organized and be shortened. It may be easier for the readers if the authors define properly the mixture of regression model and the class‐ membership equation first before moving to the computation of the GINI and of the Polarization of subgroups. Sections 2.1 and 2.2 are too long and can be significantly reduced. In section 2.1 the authors assume the condition uk > uj, but this does not appear anywhere else in the calculation of the mixture of regression model. After equation (10) all the other equations are not numbered.
Response 1:The acronyms used before the definition has been cancelled.
Point 2: The meaning of the transitional arc is not defined, hence not well understood
Response 2:It is more accurate to use “necking” instead of the transitional arc.
Point 3: Throghout the paper the notations of "upper" and "lower" regions should be better replaced with "top" and "bottom" regions respectively.
Response 3:In this paper, the notations of "upper" and "lower" have been replaced with "top" and "bottom" regions respectively.
Point 4: Figure 7: missing units in the legends.
Response 4:Replace Fig.7 with hardness curves at 1 mm, 4 mm and 7 mm from the top surface of the FPW joints.
Point 5: Figures 7,8: discussion and interpretation of the curves is inappropriate.
Response 5:The curves of Fig.7 and 8 are reanalyzed and discussed.
Point 6: Figure 14: the specification that the image comes from TEM analysis must be given.
Response 6:Clearer images were used and reanalyzed.
Point 7: Specification of the time passed after FPW before tensile testing and TEM inspection must be given.
Response 7:Tensile testing and TEM inspection were done within 7-10 days after FPW welding.
Point 8: Estimates on the cooling rate after FPW must be provided along with the orientative time cycle of FPW.
Response 8:The hardness curves were discussed by quoting B Du’s analysis conclusion on cooling rate of FPW joints.
Point 9: It is highly disputable the significant role ascribed to "precipitate evolution" without a fair comparison between precipitates morfology before and after FPW.
Response 9:Fig. 14 supplements the TEM images of the original plug.
Point 10: On the following statament: "In RZ, it can be found that although a large volume of θ″ phase dissolved, the initial grains transform to fine equiaxed grains and generated a large number of dislocations." is highly questionable unless clear experimental proof is given. Moreover, it is unsound that the pecipitate dissolution during severe heating in FPW is less important but overaging after welding and cooling is dominant.
Response 10:The precipitates in different regions of the FPW joint was reanalyzed.

Reviewer 2 Report
Though the article presented is novel with scientific soundness, it needs extensive editing of English language. The authors are using lots of complex sentences. Punctuation, articles etc. are not taken into account. Many sentences needs rewriting. Abstract is also to be rewritten.
To me the authors in this article want: To address "mechanical properties of Friction Plug welded 8 mm thick AA2219-T87 sheet". It is an interesting issue specially for aerospace engineering. They not only exploited the proposed properties but also verified them with the support of the studies of micro-graphs. The topic is original though previously some studies are done Friction Plug welded 10 mm thick AA2219-T87 sheet. The authors are needed to refer the article. The conclusions are well addressed on the main question with proper support. However, the article needs to rewrite the whole paper including the Abstract taking care of the English language.Author Response
Point 1: Though the article presented is novel with scientific soundness, it needs extensive editing of English language. The authors are using lots of complex sentences. Punctuation, articles etc. are not taken into account. Many sentences needs rewriting. Abstract is also to be rewritten.
To me the authors in this article want: To address "mechanical properties of Friction Plug welded 8 mm thick AA2219-T87 sheet". It is an interesting issue specially for aerospace engineering. They not only exploited the proposed properties but also verified them with the support of the studies of micro-graphs. The topic is original though previously some studies are done Friction Plug welded 10 mm thick AA2219-T87 sheet. The authors are needed to refer the article. The conclusions are well addressed on the main question with proper support. However, the article needs to rewrite the whole paper including the Abstract taking care of the English language.
Response 1:The whole article have been rewritten.

Reviewer 3 Report
Dear Authors.
The paper with the title “ "Characteristics of friction plug joints for AA2219-T87 FSW Welds" presents interesting and original results. However, it needs to be revised to improve the quality of presentation and understanding in the aspects of:
1. English: There are still writing errors that need to be corrected with a native speaker english proofreader.
2. The introduction needs to sharpen the originality of the paper by comparing it with related papers especially with the authors previous papers, research gaps and the reasons for doing this research.
3. Method: detailed research procedures from the preparation of the FSW specimen to the FPW process need to be written down. The forging force in the FPW is not clearly controlled.
4. Result and discussion: Some of the pictures are not clear and the discussion section does not explain the reasons why the tensile strength of the FPW joint is found in certain rpm and welding force parameters by using all data that presented in the paper.
5. Conclusion: Need improvement by adding brief reasons why the joint type and parameter rpm and certain welding force obtained maximum tensile strength approaching the strength of the base metal.
6. More detailed comments and questions can be seen in the attached pdf file. Please highlight the revision results in light green so I can review again.
Kind regards,

Author Response
Point 1: English: There are still writing errors that need to be corrected with a native speaker english proofreader.
Response 1:Corrected the language expressions of the whole article.
Point 2: The introduction needs to sharpen the originality of the paper by comparing it with related papers especially with the authors previous papers, research gaps and the reasons for doing this research
Response 2:In the introduction, the comparison with related papers is added, and the research content and innovation of this paper are introduced.
Point 3: Throghout the paper the notations of "upper" and "lower" regions should be better replaced with "top" and "bottom" regions respectively.
Response 3:Method: detailed research procedures from the preparation of the FSW specimen to the FPW process need to be written down. The forging force in the FPW is not clearly controlled.
Point 4: Result and discussion: Some of the pictures are not clear and the discussion section does not explain the reasons why the tensile strength of the FPW joint is found in certain rpm and welding force parameters by using all data that presented in the paper.
Response 4:The unclear figures have been adjusted. In the conclusion, the mechanical properties of the FPW joints were re-discussed.
Point 5: Conclusion: Need improvement by adding brief reasons why the joint type and parameter rpm and certain welding force obtained maximum tensile strength approaching the strength of the base metal.
Response 5:The conclusions have been re-extracted.
Point 6: More detailed comments and questions can be seen in the attached pdf file. Please highlight the revision results in light green so I can review again.
Response 6:All modifications are highlighted in green. In Fig.4, the Type D joint have been cut off the whole flash and the Macroappearance with the top flash cannot be obtained.

Reviewer 4 Report
Comments-
-Abstract- Please rewrite the abstract. It must be such that reader have clear understanding of the paper in brief. A brief statement (two or three line) is required in abstract section by considering a quantitative changes of Characteristics (properties) such as % change in strength with respect to the process parameter.
-Introduction
- First line- “2219 aluminum alloy is Al-Cu-Mn alloy, …………………. corrosion resistance”. The sentence is not clear please rewrite.
Experiment
- - First line- “The measured 0.2 % offset yield strength (YS), Ultimate …………………… and 12 %, respectively”. The sentence is not clear please rewrite.
- What is room temperature and low temperature. It is missing and its very important.
- What will be the diameter at the tip of plug and minimum diameter of the hole when it tapered.
- What will be the speed of plug advancement towards the hole.
Results and discussion
-Many figures have less height of text or visibility of scale text so that it is not clearly visible. Such as Fig.7(a)- Increase text (font) height in the figure.
-There are lots of grammatical mistakes and unclear sentences.
Conclusion-
- This part has to be re-written by considering following things-
o Instead of high or low please provide the appropriate quantity such as “high tensile properties- quantify it”.
o There are lots of grammatical mistakes and unclear sentences.
Author Response
Point 1: Abstract- Please rewrite the abstract. It must be such that reader have clear understanding of the paper in brief. A brief statement (two or three line) is required in abstract section by considering a quantitative changes of Characteristics (properties) such as % change in strength with respect to the process parameter.
Response 1:The abstract has been rewritten.
Point 2: First line- “2219 aluminum alloy is Al-Cu-Mn alloy, …………………. corrosion resistance”. The sentence is not clear please rewrite.
Response 2:The sentence have been rewritten.
Point 3: Experiment- First line- “The measured 0.2 % offset yield strength (YS), Ultimate …………………… and 12 %, respectively”. The sentence is not clear please rewrite.
Response 3:The sentence have been rewritten.
Point 4: What is room temperature and low temperature. It is missing and its very important.
Response 4:The Room temperature and low temperature refer to temperatures of 298 K and 77 K, respectively. Fig. 17 shows the tensile properties of the base metal, FSW joints and FPW joints for FSW weld at 298 K and 77 K.
Point 5: What will be the diameter at the tip of plug and minimum diameter of the hole when it tapered.
Response 5:The diameter at the top of the plug and the minimum diameter of the hole when it tapered are not marked, because they can be calculated from known data.
Point 6: What will be the speed of plug advancement tos the hole.
Response 6:The table 2 added welding speed: 50 mm/min.
Point 7: Results and discussion-Many figures have less height of text or visibility of scale text so that it is not clearly visible. Such as Fig.7(a)- Increase text (font) height in the figure.
Response 7:The unclear figures have been adjusted.
Point 8: There are lots of grammatical mistakes and unclear sentences.
Response 8:Corrected the language expressions of the whole article.
Point 9: Conclusion--This part has to be re-written by considering following things-
Instead of high or low please provide the appropriate quantity such as “high tensile properties- quantify it”.There are lots of grammatical mistakes and unclear sentences.
Response 9:The conclusions have been re-extracted.

Round 2
Reviewer 3 Report
Dear Authors,
Thank you very much for revising your paper with the title "Characteristics of friction plug joints for AA2219-T87 FSW Welds" and trying to fulfill my comments. There are several comments that can be followed up so that the paper is presented better as follows:
- Figure 1(a) is slightly bigger than (b),(c),(d) even with the same dimension of 20 mm diameter. It is better to make the same size in the Fig.1.
- Please explain the process of FPW that has been performed to describe Figure 2. Please write the caption for each figure in Figure 2 and explain it in the paragraph.
- What does dashed line mean in the second Figure 2? Is it FPW joint location? Please write a note in the Figure 2(b),(c).
- There is no Figure 2b and 2c only Figure 2 that exists, and there are two Figure 2. Please revise to be correct number of figures.
- Why did the D4 parameter use in this study? please write the reason.
- The Figure 7 and 8 are still too small, the size of fonts it should be the same with the font size of the paragraph.
- It is better to show the zones in a diagram of FPW joints, to make more understandable in 3.4.2.
- It should be references to show that the precipitates in the Figure 14 is theta or theta'.
- Please mention what FPW joint that was discussed in 3.4.5 Local tensile property.
- Comments and questions can be seen in the attached pdf file. Please highlight the revision results in yellow so I can check.
- I have checked the similarity of this paper using Turnitin and got a score of 22% similarity (attached file). Please revise to meet the publication standards in MDPI - Materials, if necessary. Please consult with the editors.
Kind regards,

Author Response
Figure 1(a) is slightly bigger than (b),(c),(d) even with the same dimension of 20 mm diameter. It is better to make the same size in the Fig.1.
Reply: The plug in Fig. 1a has been redrawn.
Please explain the process of FPW that has been performed to describe Figure 2. Please write the caption for each figure in Figure 2 and explain it in the paragraph.
Reply: The FPW process has been explained and added the caption for each figure in Fig. 2
What does dashed line mean in the second Figure 2? Is it FPW joint location? Please write a note in the Figure 2(b),(c).
Reply: The dashed line represents the binding interface, which has been annotated in Fig. 2b and c.
There is no Figure 2b and 2c only Figure 2 that exists, and there are two Figure 2. Please revise to be correct number of figures.
Reply: Changed one of Fig. 2 to Fig. 3.
Why did the D4 parameter use in this study? please write the reason.
Reply: In 3.4.1, FPW for FSW welds were performed using Type C plug and plate hole combination and D4 welding parameters due to defect-free weld formation and higher mechanical properties.
The Figure 7 and 8 are still too small, the size of fonts it should be the same with the font size of the paragraph.
Reply: In Fig. 7 and 8, the size of fonts have been adjusted.
It is better to show the zones in a diagram of FPW joints, to make more understandable in 3.4.2.
Reply: Different zones are marked in Fig. 12 and 13.
It should be references to show that the precipitates in the Figure 14 is theta or theta'.
Reply: Tt is concluded that the precipitates in different regions of the FPW joints are θ and θ' referring to Fig. 14k and l.
Please mention what FPW joint that was discussed in 3.4.5 Local tensile property.
Reply: In 3.4.5, it has been stated that FPW joints were obtained at D4 parameters.
Comments and questions can be seen in the attached pdf file. Please highlight the revision results in yellow so I can check.
Reply: The revision results have been highlighted in yellow.
I have checked the similarity of this paper using Turnitin and got a score of 22% similarity (attached file). Please revise to meet the publication standards in MDPI - Materials, if necessary. Please consult with the editors.
Reply: The similar sentences checked in this paper have been modified.
Checked reference [5], the author used fricture not friction:
[5] B. Du, X. Yang, K. Liu, et al, Weakening mechanism and tensile fracture behavior of AA 2219-T87 fricture plug welds, Mater. 59 Sci. Eng. A. 693(2017)129-135.
Reviewer 4 Report
How you able to test tensile properties for micro as well as macro at 77 K. Is there specific attachment or machine facility. Please provide the image of setup in comment section.
Author Response
How you able to test tensile properties for micro as well as macro at 77 K. Is there specific attachment or machine facility. Please provide the image of setup in comment section.
Reply: low temperature (77 K) tensile properties of the FPW joints were tested on the CSS-44100 universal testing machine with a incubator.